# Mechanically activated ion channel Piezo1 modulates macrophage polarization and stiffness sensing

Hamza Atcha[1,2], Amit Jairaman[3], Jesse R. Holt [3,4], Vijaykumar S. Meli[1,2,5], Raji R. Nagalla [1,2], Praveen Krishna Veerasubramanian [1,2], Kyle T. Brumm[1,2], Huy E. Lim[1,2], Shivashankar Othy[3], Michael D. Cahalan[3], Medha M. Pathak [1,3,4] & Wendy F. Liu [1,2,5]✉

Macrophages perform diverse functions within tissues during immune responses to pathogens and injury, but molecular mechanisms by which physical properties of the tissue regulate macrophage behavior are less well understood. Here, we examine the role of the mechanically activated cation channel Piezo1 in macrophage polarization and sensing of microenvironmental stiffness. We show that macrophages lacking Piezo1 exhibit reduced inflammation and enhanced wound healing responses. Additionally, macrophages expressing the transgenic $Ca^{2+}$ reporter, Salsa6f, reveal that $Ca^{2+}$ influx is dependent on Piezo1, modulated by soluble signals, and enhanced on stiff substrates. Furthermore, stiffness-dependent changes in macrophage function, both in vitro and in response to subcutaneous implantation of biomaterials in vivo, require Piezo1. Finally, we show that positive feedback between Piezo1 and actin drives macrophage activation. Together, our studies reveal that Piezo1 is a mechanosensor of stiffness in macrophages, and that its activity modulates polarization responses.

[1] Department of Biomedical Engineering, University of California Irvine, Irvine, USA. [2] The Edwards Lifesciences Center for Advanced Cardiovascular Technology, University of California Irvine, Irvine, USA. [3] Department of Physiology and Biophysics, University of California Irvine, Irvine, USA. [4] Sue and Bill Gross Stem Cell Research Center, University of California Irvine, Irvine, USA. [5] Department of Chemical and Biomolecular Engineering, University of California Irvine, Irvine, USA. ✉email: wendy.liu@uci.edu

Macrophages are versatile cells of the innate immune system that play critical roles in homeostasis, pathogen defense, and response to injury. This diversity in macrophage function results from their ability to respond dynamically to cues in their microenvironment and polarize towards a multitude of functional phenotypes[1–3]. Pathogens and tissue damage polarize macrophages towards a "classically activated" state that promotes inflammation, whereas wound healing cytokines lead to an "alternatively activated", anti-inflammatory state that aids in tissue repair[4]. In addition to such soluble stimuli, physical cues including tissue stiffness, matrix architecture, and mechanical stimulation are also thought to contribute to macrophage function[5–8]. Our group and others have shown that macrophages cultured on soft substrates (~Pa–kPa) have reduced inflammatory activation when compared to cells adhered to glass or other stiff substrates (~kPa–GPa)[5,9,10]. In addition, mechanical mismatch that occurs when stiff surgical implants are placed within soft tissues has been shown to enhance immune cell recruitment and inflammatory activation[11,12]. Inflammation caused by stiff materials is also associated with a more severe foreign body response and thicker fibrous capsule formation when compared to soft implants, suggesting significant pathological consequences of stiffness-enhanced inflammation[10]. Moreover, increases in tissue stiffness are characteristic of numerous diseases in which macrophages are involved[10,13,14]. While stiffness clearly affects macrophage function, the molecular mechanisms underlying macrophage stiffness sensing are still poorly understood.

Stretch-activated ion channels allow the passage of ions in response to increased membrane tension and play a crucial role in the detection and transduction of external physical stimuli into electrochemical activity that influences signaling and cell behavior[15,16]. A recent flurry of research shows that the mechanically activated, non-specific cation channel Piezo1 in particular is involved in numerous developmental processes and pathological conditions[17–21] and it transduces a variety of mechanical cues[22–24]. While increased $Ca^{2+}$ influx through transient receptor potential (TRP) channels, including TRPM7 and TRPC1, is widely recognized to contribute to macrophage activation by inflammatory agonists[25,26], the role of Piezo1 and the $Ca^{2+}$ signals it generates in response to soluble stimuli remains less well explored. Recent studies have identified Piezo1 as a mechanosensor of pressure and shear stress in myeloid cells recruited to the lung, heart, and tumors, and found that channel activity stimulates inflammation[27–29]. However, macrophages in most tissues are not subjected to such extreme mechanical stresses, and the mechanical cue that all macrophages are likely to encounter in their environment is variations in matrix stiffness. While Piezo1 has been shown to sense stiffness in neuronal stem cells and glial cells[17,18], a role in macrophage stiffness sensing has not yet been described.

In this study, we investigate the role of Piezo1 and $Ca^{2+}$ influx in regulating macrophage responses within varying stiffness environments. We find that Piezo1 activity promotes interferon-γ (IFNγ) and lipopolysaccharide (LPS)-induced inflammatory and suppresses interleukin-4 (IL4) and interleukin-13 (IL13)-induced healing responses. Macrophages expressing the transgenic $Ca^{2+}$ reporter, Salsa6f, reveal that Piezo1-dependent $Ca^{2+}$ influx is modulated by soluble signals and increases on stiff substrates. Furthermore, Piezo1 regulates both stiffness-dependent changes in macrophage function in vitro and modulates the immune response to subcutaneous implantation of biomaterials in vivo. Finally, we show that positive feedback between Piezo1 and actin drives macrophage activation.

## Results

### Piezo1 modulates macrophage inflammatory and healing responses.
We first evaluated the expression of Piezo1 in murine bone marrow-derived macrophages (BMDMs) from wild type C57BL/6 J mice, and compared it to the expression of several other channels with known functions in macrophages[25,26,30–33]. Piezo1 was the most abundantly expressed channel and was more highly expressed than its homolog Piezo2, confirming a recent report (Supplementary Figs. 1 and 2)[27]. In addition, Piezo1 expression increased following stimulation by potent inflammatory agonists, IFNγ and LPS (Supplementary Fig. 2). To further explore the role of Piezo1 in regulating macrophage function, we generated Piezo1$^{flox/flox}$LysM$^{Cre/+}$ mice with Piezo1 conditionally depleted from LysM-expressing myeloid cells (Piezo1$^{ΔLysM}$) and confirmed reduction of Piezo1 gene expression in isolated BMDMs (Supplementary Fig. 3). We then evaluated the expression of prototypical inflammatory and healing markers in response to in vitro stimulation. We found that BMDMs harvested from Piezo1$^{ΔLysM}$ mice have significantly reduced expression of the inflammatory marker inducible nitric oxide synthase (iNOS) in response to IFNγ/LPS stimulation when compared to BMDMs isolated from control Piezo1$^{flox/+}$LysM$^{Cre/+}$ (Piezo1$^{fl/+}$) mice (Fig. 1a). In contrast, Piezo1$^{ΔLysM}$ BMDMs exhibited a significant increase in the expression of the healing marker arginase-1 (ARG1) in response to wound healing cytokines IL4/IL13, when compared to controls (Fig. 1a). iNOS or ARG1 expression was not observed in unstimulated macrophages, suggesting that Piezo1 expression alone does not polarize macrophages, but it modulates activation by external soluble signals. Examination of cell morphology, which we have previously shown to correlate with activation state[34], showed that Piezo1$^{ΔLysM}$ BMDMs were more highly elongated compared to control BMDMs in both unstimulated and IFNγ/LPS stimulated conditions, whereas cells had similar shape in IL4/IL13 stimulated conditions (Supplementary Fig. 3). Analysis of other markers of activation, both secreted proteins and gene expression, also showed reduced inflammation and increased healing responses in macrophages lacking Piezo1. Piezo1$^{ΔLysM}$ BMDMs secreted significantly less TNFα and IL6 and had lower Il6 and Nos2 gene expression upon stimulation with IFNγ/LPS when compared to control Piezo1$^{fl/+}$ macrophages (Fig. 1b, c). In addition, IL4/IL13-induced Arg1 and Retnla gene expression was higher in Piezo1$^{ΔLysM}$ compared to control BMDMs (Fig. 1c). Resident macrophages isolated from the peritoneum of Piezo1$^{ΔLysM}$ mice also had suppressed iNOS and enhanced ARG1 expression, similar to what was observed with BMDMs (Supplementary Fig. 4). Together, these studies suggest that Piezo1 expression enhances inflammatory and reduces wound healing responses in macrophages.

To probe potential downstream mechanisms underlying Piezo1-mediated effects, we examined the role of Piezo1 in regulating transcription factors involved in macrophage activation. Nuclear factor kappa-light-chain-enhancer of activated B cells (NFκB) and signal transducer and activator of transcription 6 (STAT6) are essential transcription factors in inflammatory and healing activation, respectively[35–42]. In addition, the activation of both transcription factors is known to be dependent on $Ca^{2+}$, where increased intracellular $Ca^{2+}$ has been shown to enhance NFκB and dampen STAT6 activation[25,43]. We found that cells lacking Piezo1 exhibited reduced IFNγ/LPS-induced NFκB activation when compared to control cells (Fig. 1d). IFNγ/LPS stimulated Piezo1$^{ΔLysM}$ BMDMs also had reduced activation and nuclear localization of STAT1 and STAT3, transcription factors known to play a role in inflammation (Supplementary Fig. 5). On the other hand, IL4/IL13-stimulated BMDMs deficient in Piezo1 showed increased expression of p-STAT6 (Fig. 1e). These results are consistent with the reduced IFNγ/LPS-induced inflammatory and enhanced IL4/IL13-induced healing activation observed in Piezo1$^{ΔLysM}$ BMDMs (Fig. 1a–c). Together, these studies suggest that transcriptional regulation of macrophage inflammatory and healing responses are modulated by Piezo1.

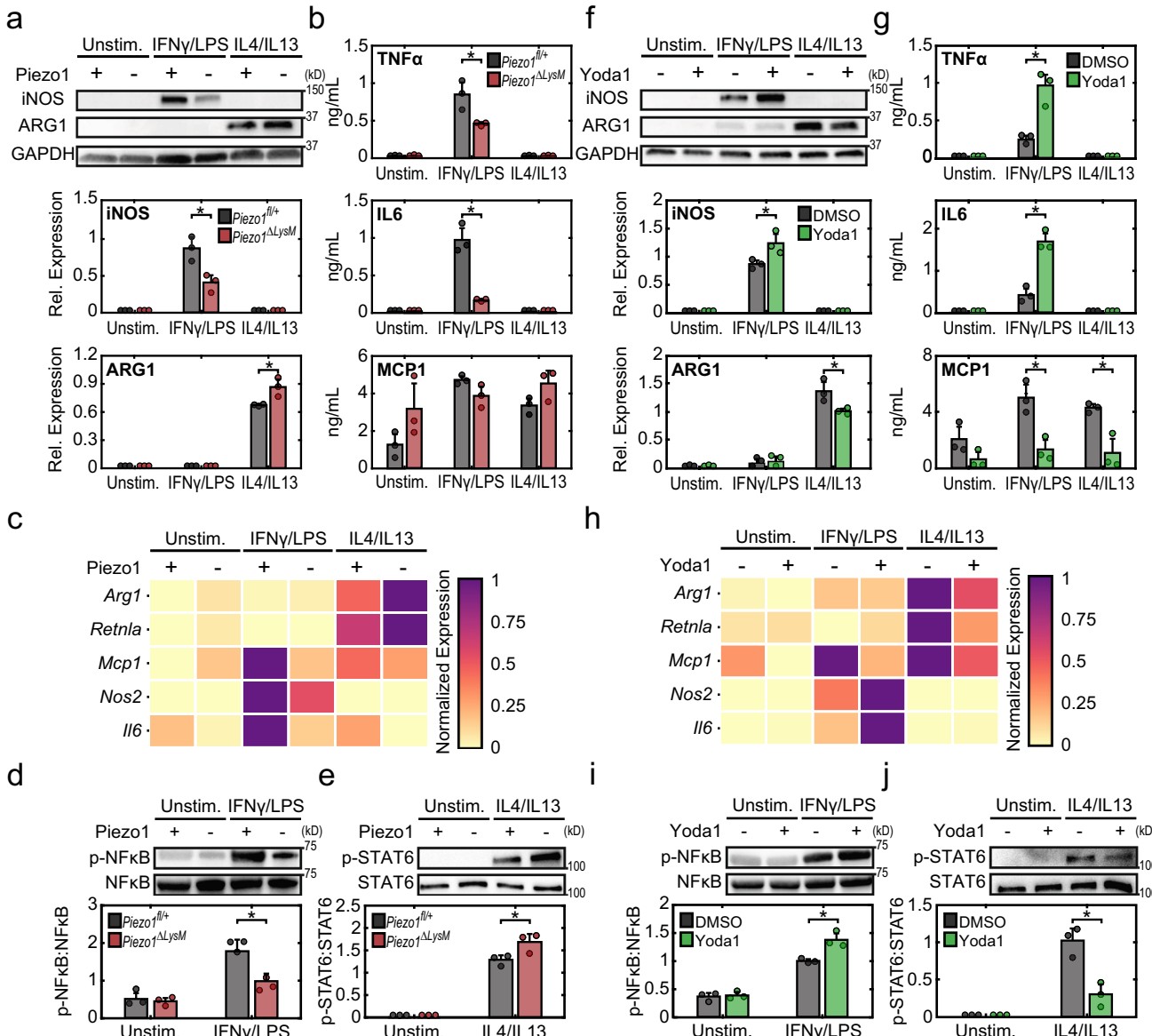

**Fig. 1 Piezo1 regulates cytokine-induced macrophage activation. a** Representative Western blots (top) and quantification (bottom) of iNOS, ARG1, and GAPDH of *Piezo1fl/+* and *Piezo1ΔLysM* BMDMs incubated with media (Unstim.), IFNγ/LPS (0.3 ng/mL of each), or IL4/IL13 (0.1 ng/mL of each). **b** TNFα, IL6, and MCP1 secretion from *Piezo1fl/+* and *Piezo1ΔLysM* BMDMs incubated with the indicated conditions, as measured by ELISA. **c** Relative gene expression of inflammatory and healing markers in *Piezo1fl/+* and *Piezo1ΔLysM* BMDMs incubated with the indicated conditions, as measured by qPCR. Gene expression is shown relative to the highest expressing condition. **d, e** Representative Western blots (top) and quantification (bottom) of p-NFκB/NFκB of *Piezo1fl/+* and *Piezo1ΔLysM* BMDMs incubated with media (Unstim.) or IFNγ/LPS for 1 h (**d**) and p-STAT6/STAT6 of *Piezo1fl/+* and *Piezo1ΔLysM* BMDMs incubated with media (Unstim.) or IL4/IL13 for a period of 1 h (**e**). **f** Representative Western blots (top) and quantification (bottom) of iNOS, ARG1, and GAPDH of BMDMs exposed to DMSO or 5 μM Yoda1 and stimulated with media (Unstim.), IFNγ/LPS (0.3 ng/mL of each), or IL4/IL13 (0.1 ng/mL of each). **g** TNFα, IL6, and MCP1 secretion from BMDMs exposed to DMSO or 5 μM Yoda1 and incubated in the indicated conditions, as measured by ELISA. **h** Relative gene expression of inflammatory and healing markers of BMDMs exposed to DMSO or 5 μM Yoda1 incubated in the indicated conditions, as measured by qPCR. Gene expression is shown relative to the highest expressing condition. **i–j** Representative Western blots (top) and quantification (bottom) of p-NFκB/NFκB of wild-type BMDMs exposed to DMSO or Yoda1 and stimulated with media (Unstim.) or IFNγ/LPS for 2 h (**d**) and p-STAT6/STAT6 of *Piezo1fl/+* and *Piezo1ΔLysM* BMDMs incubated with media (Unstim.) or IL4/IL13 for a period of 1 h. Error bars denote Mean ± SD for three independent experiments, * *p* < 0.05 as determined by two-tailed Student's *t* test. Phosphorylated and total forms of transcription factors were obtained from loading equal amounts of protein in separate gels and the resulting blots were processed in parallel. Source data including exact p-values are provided as a Source Data file.

To examine the effects of other modulators of Piezo1 and ion channel activity, we used siRNA to knock down Piezo1 expression, gadolinium chloride (GdCl₃) to broadly target ion channels, and GsMTx-4 to target mechanosensitive ion channels. We observed that Piezo1 siRNA, GdCl₃, and GsMTx-4 all generally suppressed both inflammatory responses to IFNγ/LPS and healing responses to IL4/IL13 in BMDMs (Supplementary

Figs. 6, 7), confirming the role of Piezo1 in inflammation but suggesting that constitutive knockout of this ion channel may be required for upregulating healing responses, and that transient or incomplete siRNA-mediated knockdown as well as non-specific channel inhibition with GdCl₃ or GsMTx-4 may not be sufficient. Conversely, to evaluate whether activating Piezo1 influences macrophage function, we utilized a Piezo1-specific agonist,

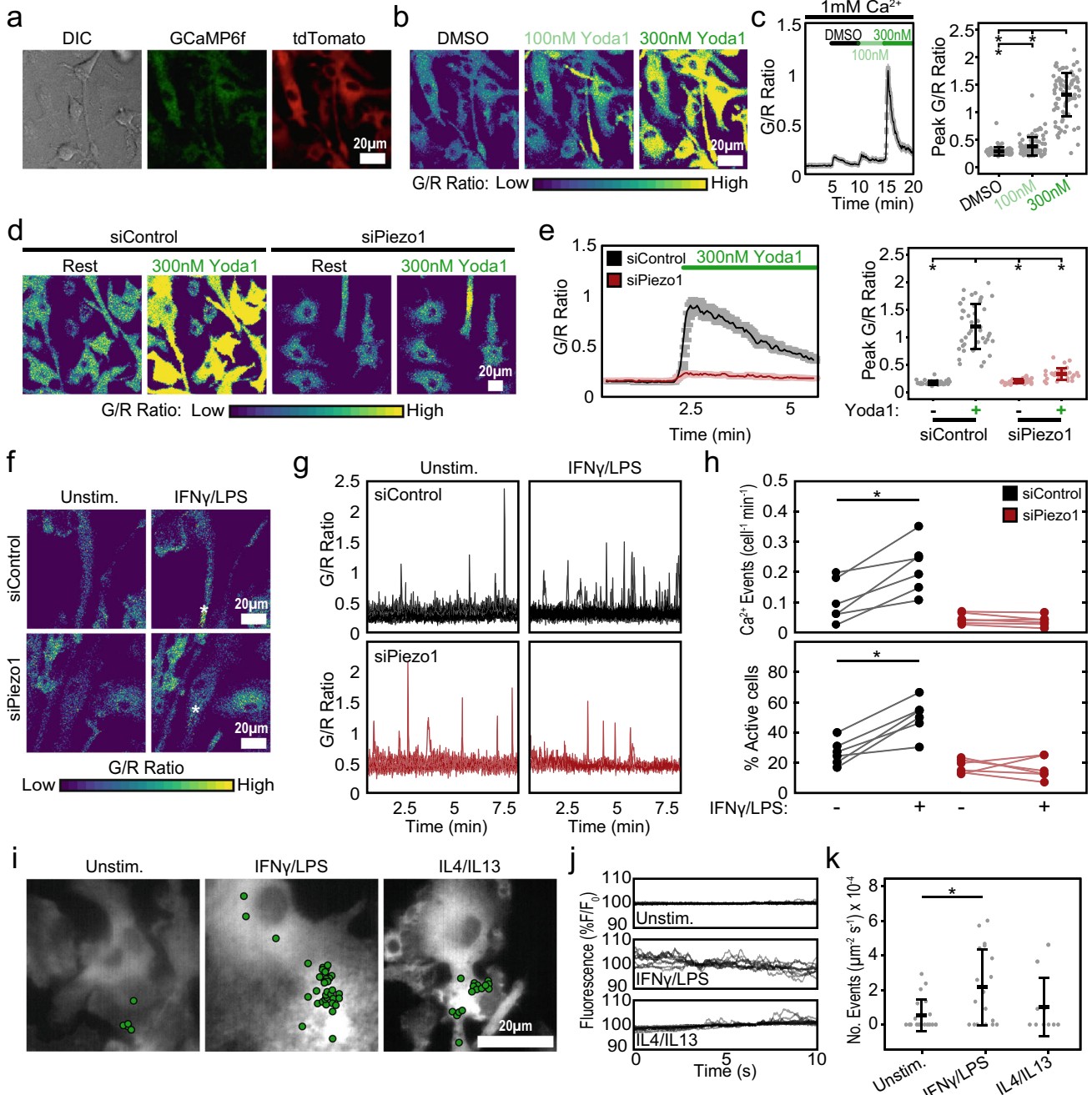

Yoda1[44]. We observed that Yoda1 treatment increased expression of iNOS, secretion of TNFα and IL6, and expression of *Il6* and *Nos2* genes in response to IFNγ/LPS (Fig. 1f–h). In contrast, Yoda1 treatment reduced expression of ARG1 protein and *Arg1* and *Retnla* genes in response to IL4/IL13. We also evaluated changes in NFκB and STAT6 activation following treatment with Yoda1 and found that Yoda1 enhanced NFκB and suppressed STAT6 activation in IFNγ/LPS and IL4/IL13 treated BMDMs, respectively (Fig. 1i, j). Together with the studies in *Piezo1ΔLysM* BMDMs, these data suggest that Piezo1 activity enhances NFκB-mediated inflammatory activation and may also inhibit STAT6-mediated healing activation in macrophages.

**Piezo1 modulates Ca²⁺ influx in macrophages.** Previous studies have shown that Ca²⁺ signals play a critical role in promoting

macrophage inflammatory activation[25,26,31]. Piezo1 is a non-selective ion channel that regulates the influx of several cations including Ca²⁺; however, its role in regulating macrophage Ca²⁺ signals is unknown. To address this gap in knowledge, we utilized the Salsa6f probe to visualize Ca²⁺ signals in BMDMs. Salsa6f is a genetically encoded ratiometric Ca²⁺ indicator, consisting of tdTomato fused to the Ca²⁺ indicator GCaMP6f by a V5 epitope tag (tdTomato-V5-GCaMP6f)[45,46]. Calculation of single-cell GCaMP6f/tdTomato (G/R) ratio enables quantification of cytosolic Ca²⁺ levels independent of probe concentration or cell movement. Homozygous LSL-Salsa6f mice were crossed with Vav1Cre/Cre mice to selectively express Salsa6f in all hematopoietic cells[45,46], including macrophages. As expected, Salsa6f protein was expressed in the cytosol and excluded from the nucleus in BMDMs (Fig. 2a). We treated Salsa6f+ BMDMs with Yoda1 in the presence of extracellular Ca²⁺ to confirm the

**Fig. 2 Regulation of Ca$^{2+}$ influx by Piezo1 channels in macrophages. a** Representative differential interference contrast (DIC) and fluorescence images showing expression of Salsa6f probe in BMDMs isolated from Vav1-Salsa6f mice, tdTomato is displayed in red and GCaMP6f is displayed in green. **b** Green:Red (G/R) ratio images showing Ca$^{2+}$ responses to Yoda1 in Salsa6f+ BMDMs when exposed to DMSO (black), 100 nM (light green), and 300 nM (dark green) Yoda1, all in 1 mM Ca$^{2+}$ Ringer solution. **c** G/R traces averaged across all cells in a field of view over time (left) and quantification of peak G/R ratios per cell (right) of Salsa6f+ BMDMs in response to Yoda1. $N = 87$ cells, representative of three independent experiments, error bars denote Mean ± SD, * $p < 0.05$ as determined by two-tailed Mann–Whitney $U$ test). **d** Representative images showing Ca$^{2+}$ responses to 300 nM Yoda in Salsa6f+ BMDMs treated with either non-target (siControl) or Piezo1 (siPiezo1) siRNA. **e** G/R traces averaged across all cells in a field of view over time (left) and quantification of peak G/R intensities per cell (right) of siControl and siPiezo1 treated BMDMs exposed to Ringer solution (Rest) or 300 nM Yoda1 in Ringer solution. $N = 46$ and 31 cells for siControl and siPiezo1 conditions, representative of three independent experiments, error bars denote Mean ± SD, * $p < 0.05$ as determined by two-tailed Mann–Whitney U test). **f–h** Representative G/R ratio images (**f**), traces of individual Ca$^{2+}$ events (**g**), and quantification of number of Ca$^{2+}$ events (normalized for cell number and time) and fraction of cells showing Ca$^{2+}$ elevations (**h**) taken from a time-lapse video of siControl and siPiezo1 treated Salsa6f+ BMDMs following acute addition of Ringer solution (Unstim.) or Ringer solution containing 100 ng/mL IFNγ/LPS. Asterisks denote the occurrence of a Ca$^{2+}$ event. Each data point in (**h**) denotes a single video ($N = 6$ videos, * $p < 0.05$ as determined by two-tailed paired $t$ test). **i–k** Representative images overlaid with centroids denoting Ca$^{2+}$ flickers in green (**i**) and traces of individual Ca$^{2+}$ flickers (**j**) recorded in unstimulated, 0.3 ng/mL IFNγ/LPS, and 0.1 ng/mL IL4/IL13 stimulated Salsa6f+ BMDMs using high speed TIRF microscopy. **k** Frequency of Ca$^{2+}$ flickers in unstimulated, IFNγ/LPS, and IL4/IL13 stimulated BMDMs. Each data point represents the frequency of Ca$^{2+}$ flickers in a single video each composed of one or more cells. $N = 21$, 18, and 10 fields of view for Unstim., IFNγ/LPS and IL4/IL13 conditions, respectively. Error bars denote Mean ± SD, and * $p < 0.05$ as determined by two-tailed Mann–Whitney $U$ test). Source data including exact p-values are provided as a Source Data file.

presence of functional Piezo1 channels on the cell membrane. Yoda1 produced a dose-dependent increase in cytosolic Ca$^{2+}$ measured as changes in single-cell G/R ratios (Fig. 2b, c). Moreover, depletion of Piezo1 by siRNA significantly inhibited Yoda1-induced Ca$^{2+}$ influx, confirming the specificity of Yoda1 in targeting Piezo1 while also validating reduced ion channel expression (Fig. 2d–e and Supplementary Video 1).

We next determined the role of Piezo1 in regulating Ca$^{2+}$ events in response to cytokine-mediated macrophage activation. We first imaged Salsa6f+ BMDMs as they were acutely exposed to Ringer solution-alone (control), and then containing IFNγ/LPS or IL4/IL13, allowing comparison of cytosolic Ca$^{2+}$ responses between control and soluble stimuli conditions within the same field of view. Under these conditions, we observed both local and global elevations in cytosolic Ca$^{2+}$ (Supplementary Fig. 8). More importantly, we found that stimulation with inflammatory cytokines (IFNγ/LPS) led to greater Ca$^{2+}$ activity, as indicated by increased number of Ca$^{2+}$ events and percentage of cells actively showing Ca$^{2+}$ influx (Supplementary Fig. 8, see methods section for details on quantification). No increases were observed with acute IL4/IL13 treatment, suggesting that increased Ca$^{2+}$ activity may be a unique response to inflammatory stimuli (Supplementary Fig. 8). To determine the specific role of Piezo1 in this process, Salsa6f+ BMDMs were treated with control non-target (siControl) or Piezo1 siRNA (siPiezo1). siPiezo1 treatment prevented the increase in the number of Ca$^{2+}$ events observed in response to acute addition of IFNγ/LPS, which was maintained in siControl-treated BMDMs (Fig. 2f–h and Supplementary Video 2). These data establish a role for Piezo1 in mediating macrophage Ca$^{2+}$ influx in response to acute inflammatory stimuli. In addition, we also examined changes in Ca$^{2+}$ events in response to long-term exposure to inflammatory and healing activation stimuli. Imaging at a millisecond-scale temporal resolution using total internal reflection fluorescence (TIRF) microscopy[17,47] revealed significantly higher Ca$^{2+}$ activity at the membrane in inflammatory macrophages (Fig. 2i–k and Supplementary Video 3). These TIRF data show that local and transient Ca$^{2+}$ events that lasted milliseconds in scale, which fall below the resolution of confocal microscopy, are prominent in inflammatory activated macrophages. Together, these results establish a role for Piezo1 in mediating Ca$^{2+}$ signals in response to soluble inflammatory stimuli.

**Stiffness-mediated macrophage activation is dependent on Piezo1.** Substrate rigidity has been shown by our group and

others to influence macrophage inflammatory activation in response to soluble stimuli[5,9,10]. Given that the Piezo1 channel is involved in mechanosensation in many cell types and its activity has previously been shown to be modulated by substrate stiffness[17,18], we next investigated the role of stiffness in influencing Ca$^{2+}$ and Piezo1-mediated macrophage activation. Using fibronectin-conjugated polyacrylamide substrates of varying stiffness (1, 20, 40, and 280 kPa), we first evaluated the role of substrate stiffness on the expression of iNOS and ARG1 following IFNγ/LPS and IL4/IL13 treatment, respectively. We found that increases in substrate stiffness are associated with enhanced iNOS expression in response to IFNγ/LPS stimulation, with significant increases in cells on 280 kPa compared to 20 or 40 kPa (Fig. 3a). These data are consistent with earlier reports[5,9,10]. In addition, we found that macrophages generally have increased IL4/IL13-induced ARG1 expression when the stiffness of the substrate increases, with a significant difference between cells on 40 and 280 kPa (Fig. 3a). Consistent with these findings, we found that stiff substrates enhance IFNγ/LPS-induced NFκB and IL4/IL13-induced STAT6 activation (Fig. 3b–c). Inhibition of STAT6 using AS1517499 suppressed *Arg1* expression in response to IL4/IL13 stimulation in cells seeded on both 1 and 280 kPa surfaces, suggesting that STAT6 mediated healing activation is present even on soft substrates (Supplementary Fig. 9). Together, these data support the involvement of NFκB and STAT6 in stiffness-dependent macrophage activity.

We next evaluated whether Piezo1 contributes to the observed effects on inflammatory and healing activation of BMDMs cultured on different stiffness substrates. To assess Piezo1 expression, we utilized BMDMs from *Piezo1$^{P1-tdT}$* mice which express a Piezo1-tdTomato fusion protein that labels endogenous Piezo1 channels with tdTomato[21]. We found that Piezo1 expression increased in macrophages cultured on stiff (280 kPa) compared to soft (1 kPa) substrates using both immunofluorescence imaging of tdTomato and Western blot of Piezo1 (Fig. 3d and Supplementary Fig. 10). We also examined Ca$^{2+}$ influx in Salsa6f+ BMDMs seeded on different stiffness substrates. BMDMs seeded on stiffer substrates displayed more frequent Ca$^{2+}$ events with a larger percentage of cells actively involved in Ca$^{2+}$ influx following exposure to IFNγ/LPS when compared to cells cultured on soft substrates, suggesting that stiffness-dependent inflammatory activation is associated with increased Ca$^{2+}$ activity (Fig. 3e–f, Supplementary Fig. 10, and Supplementary Video 4). Consistent with these findings, Salsa6f+ BMDMs seeded on substrates of higher stiffnesses showed significantly

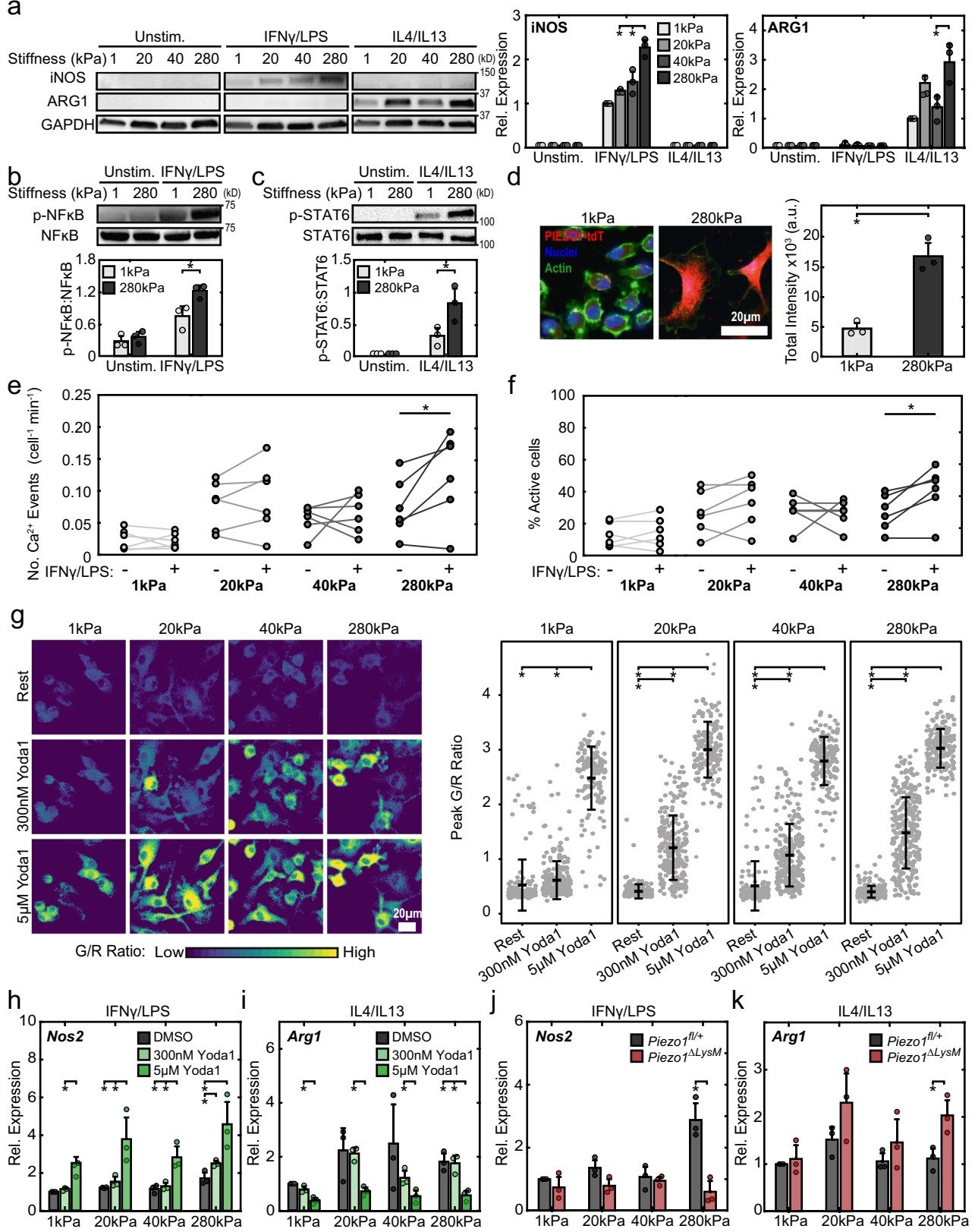

larger Ca$^{2+}$ responses to Yoda1 (300 nM) (Fig. 3g). These differences were not observed at higher doses of Yoda1 (5 μM), likely due to a maximum global increase in cytosolic Ca$^{2+}$ that obscures the differences in Piezo1 expression across different stiffnesses. Nonetheless, these results suggest that substrate stiffness increases the expression of Piezo1 at the cell membrane (Fig. 3g and Supplementary Video 5).

To probe the role of Piezo1 in stiffness-dependent changes in macrophage function, we first used Yoda1 on wild type BMDMs seeded on different stiffness substrates. We found that low doses

**Fig. 3 Stiffness-dependent Piezo1 expression/activity modulates macrophage activation. a** Representative Western blots (left) and quantification (right) of iNOS, ARG1, and GAPDH of BMDMs seeded on 1, 20, 40, and 280 kPa polyacrylamide gels and incubated with media (Unstim.), 0.3 ng/mL IFNγ/LPS, or 0.1 ng/mL IL4/IL13. Data normalized to the 1 kPa IFNγ/LPS condition. Samples were derived from the same experiment and the resulting blots were processed in parallel. **b, c** Representative Western blots (top) and quantification (bottom) of p-NFκB/NFκB (**b**) and p-STAT6/STAT6 (**c**) in wild type BMDMs seeded on 1 and 280 kPa polyacrylamide gels and stimulated with IFNγ/LPS or IL4/IL13, respectively. Phosphorylated and total forms of transcription factors were obtained from loading equal amounts of protein in separate gels and the resulting blots were processed in parallel. **d** Representative immunofluorescent images (left) and quantification of fluorescence (right) of $Piezo1^{P1-tdT}$ BMDMs seeded on polyacrylamide gels, tdTomato is displayed in red. Quantification of **e** number of $Ca^{2+}$ events (normalized for cell number and time) and **f** fraction of cells showing $Ca^{2+}$ elevations in unstimulated BMDMs seeded on 1, 20, 40, and 280 kPa polyacrylamide surfaces following acute addition of Ringer solution (Unstim.) or Ringer solution with 100 ng/mL IFNγ/LPS, captured by confocal microscopy. Each data point is calculated from a 10-min time-lapse video ($N = 6$ for each stiffness). **g** Representative G/R ratio images (left) and quantification of peak G/R intensities of BMDMs seeded on polyacrylamide gels of indicated stiffness and exposed to DMSO, 300 nM Yoda1, and 5 μM Yoda1. $N = 107, 248, 145$ cells on 1 kPa, $n = 118, 239, 169$ cells on 20 kPa, $n = 113, 250, 183$ cells on 40 kPa, and $n = 106, 274, 184$ cells on 280 Pa and exposed to Rest, 300 nM Yoda1, or 5 μM Yoda1, respectively. Data representative of three independent experiments. **h** Relative $Nos2$ gene expression of BMDMs seeded on surfaces of indicated stiffness, exposed to DMSO, 300 nM Yoda1, or 5 μM Yoda1, and stimulated with 0.3 ng/mL IFNγ/LPS. Data normalized to 1 kPa DMSO control. **i** Relative $Arg1$ gene expression of BMDMs seeded on surfaces of indicated stiffness, exposed to DMSO, 300 nM Yoda1, or 5 μM Yoda1, and stimulated with 0.1 ng/mL IL4/IL13. Data normalized to 1 kPa DMSO control. **j** Relative $Nos2$ gene expression $Piezo1^{fl/+}$ and $Piezo1^{ΔLysM}$ BMDMs seeded on surfaces of indicated stiffness and stimulated with 0.3 ng/mL IFNγ/LPS. Data normalized to 1 kPa Piezo1 control. **k** Relative $Arg1$ gene expression of $Piezo1^{fl/+}$ and $Piezo1^{ΔLysM}$ BMDMs seeded on surfaces of indicated stiffness, and stimulated with 0.1 ng/mL IL4/IL13. Data normalized to 1 kPa $Piezo1^{fl/+}$ controls. Error bars denote Mean ± SD for three separate experiments, * $p < 0.05$ as determined by two-tailed paired $t$ test (**a, e, h–k**), two-tailed Student's test (**b–d**), or Mann–Whitney $U$ test (**g**). Source data including exact p-values are provided as a Source Data file.

of Yoda1 (300 nM) significantly increased the gene expression of the inflammatory marker $Nos2$ in IFNγ/LPS-stimulated BMDMs cultured on 280 kPa hydrogels. High doses of Yoda1 (5 μM) significantly enhanced $Nos2$ expression across all stiffness substrates (Fig. 3h). High doses of Yoda1 also significantly decreased the expression of the healing marker $Arg1$ in IL4/IL13-stimulated cells (Fig. 3i). In contrast to Yoda1-induced channel activation, $Piezo1^{ΔLysM}$ BMDMs cultured on stiff substrates exhibited less $Nos2$ in response to IFNγ/LPS and increased $Arg1$ expression in response to IL4/IL13 (Fig. 3j–k). Taken together, these results support a model whereby stiffer environments are associated with increased Piezo1 expression and $Ca^{2+}$ activity, which enhances inflammation. Using a pharmacological approach, we confirmed the role of calpains, $Ca^{2+}$ dependent proteases, in regulating IFNγ/LPS induced inflammatory activation in macrophages, as has previously been shown by others[48,49] (Supplementary Fig. 11). Intriguingly, while increases in stiffness were associated with increased IL4/IL13-induced wound healing response, enhanced activation of Piezo1 by Yoda1 inhibited wound healing and inhibition of Piezo1 by genetic knockout enhanced healing, suggesting that additional mechanisms independent of Piezo1 are likely involved in stiffness-mediated enhancement of healing responses. It is possible that other ion channels may be involved, since broad inhibition of ion channels by $GdCl_3$ or GsMTx-4 resulted in reduced expression of healing markers in response to IL4/IL13 (Supplementary Figs. 6, 7). Nonetheless, these results suggest a critical role for stiffness in regulating $Ca^{2+}$ activity in macrophages and show that these effects, particularly in the context of inflammation, are dependent on Piezo1.

**Macrophage responses to different stiffness material implants in-vivo is regulated by Piezo1.** Mechanical mismatch is common between surgical implants and native tissues, and often results in poor material integration characterized by a severe foreign body response[11]. While stiff materials are often favored due to their mechanical integrity, they have also been shown to promote inflammation and are associated with thicker fibrous collagen capsule formation when compared to soft materials[11,12]. To better understand the potential role of Piezo1 in regulating the response of myeloid cells, including macrophages, during this process, we implanted precast soft (1 kPa) and stiff (140 kPa) polyethylene glycol diacrylate-400 (PEGDA-400) hydrogels subcutaneously into $Piezo1^{fl/+}$ and $Piezo1^{ΔLysM}$ mice. After 14 days of implantation, the hydrogels and surrounding tissue were harvested and examined by histology to quantify immune cell infiltration and fibrous capsule thickness (Fig. 4a–d). In control $Piezo1^{fl/+}$ mice, we found that stiff material implants significantly increased immune cell infiltration and caused a more severe foreign body response, as indicated by thicker fibrous capsule formation, when compared to soft material implants, consistent with previous reports in wild type mice[10,12]. In contrast, depletion of Piezo1 in myeloid cells abrogated these responses to stiff implants with significantly reduced immune infiltrate and thickness of the collagen capsule surrounding the implant material, comparable to soft implants. These data suggest that the expression of Piezo1 in cells of myeloid lineage plays a pivotal role in influencing the host response to different stiffness material implants.

Given that myeloid-derived macrophages are central regulators of the foreign body response, we next examined the expression of macrophage functional markers in response to implants. Tissues surrounding soft and stiff implants in $Piezo1^{fl/+}$ and $Piezo1^{ΔLysM}$ mice were harvested at 3 and 14 d after implantation, and immunohistochemistry was used to evaluate macrophage specific (F4/80+) expression of iNOS and ARG1 (Fig. 4e–f). We observed that the extent of macrophage recruitment to soft and stiff implants was similar between $Piezo1^{fl/+}$ and $Piezo1^{ΔLysM}$ mice at both 3 and 14 d after implantation (Fig. 4g). However, there were significant differences in expression of functional markers. In control $Piezo1^{fl/+}$ mice, the percentage of iNOS+ macrophages surrounding stiff implants was greater at both 3 and 14 d (Fig. 4h–i), consistent with in vitro observations (Fig. 3a). However, unlike the in vitro findings, the percentage of ARG1+ macrophages was reduced in stiff implants when compared to soft implants at 14 d in control $Piezo1^{fl/+}$ mice. These differences could potentially be attributed to the time point of evaluation. In $Piezo1^{ΔLysM}$ mice, the percentage of iNOS+ macrophages was similar and low in response to soft and stiff materials; the percentage of ARG1+ macrophages was also similar in response to soft and stiff materials at 3 d, and increased at 14 d (Fig. 4h–i). We also found significant increases in the percentage of NFκB+ macrophages within tissue surrounding stiff compared to soft implants in $Piezo1^{fl/+}$ mice, but not $Piezo1^{ΔLysM}$ mice which is consistent with our observations of reduced inflammation in

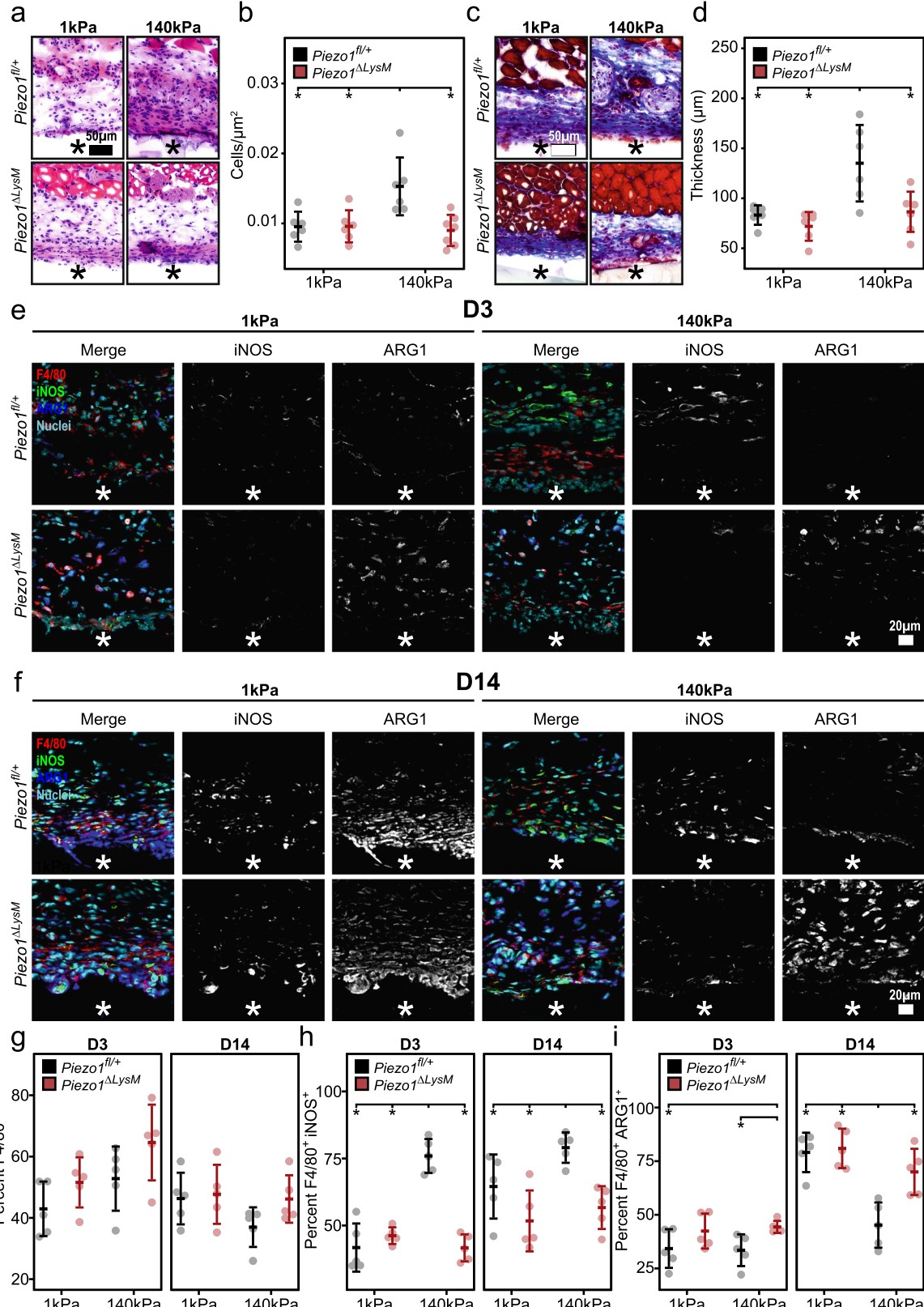

myeloid cell-specific Piezo1 deficient mice (Supplementary Fig. 12). These data again suggest that macrophages lacking Piezo1 may lose their ability to sense and respond to different stiffness environments. The decrease in inflammatory activation and increase in healing activation support the reduced immune infiltrate and fibrous capsule thickness observed in *Piezo1^{ΔLysM}* mice. Together, these results further validate the role of Piezo1 in regulating macrophage polarization responses to different stiffness environments and reveal downstream effects on fibrous capsule formation.

**Fig. 4 Piezo1 modulates the foreign body response and macrophage activation in response to stiff material implants. a–d** Representative H&E (**a**) and Masson's trichrome (**c**) stained tissue surrounding soft (1 kDa) and stiff (140 kDa) PEGDA material implanted in *Piezo1fl/+* and *Piezo1ΔLysM* mice for a period of 14 days. Quantification of immune cell infiltrate normalized to area (**b**) and average collagen capsule thickness (**d**). N = 6 for *Piezo1fl/+* mice treated with soft and stiff implants as well as *Piezo1ΔLysM* treated with soft implants, and n = 7 for *Piezo1fl/+* and *Piezo1ΔLysM* mice treated with stiff implants. **e, f** Representative immunohistochemistry images of tissue collected 3 days (D3, **e**) and 14 days (D14, **f**) post-implantation and stained for F4/80, iNOS, ARG1, and Hoechst. **g** Quantification of percent cells that stained positive for F4/80. **h** Quantification of percent cells that stained positive for F4/80 and iNOS. **i** Quantification of percent cells that stained positive for F4/80 and ARG1. Material location indicated with asterisk. N = 5 and error bars denote Mean ± SD for $n \geq 5$, * $p < 0.05$ as determined by two-tailed Student's *t* test. Source data including exact p-values are provided as a Source Data file.

**Cytoskeletal feedback regulates Piezo1-mediated Ca$^{2+}$ influx and macrophage activation.** While Piezo1 is well known for its ability to transduce external mechanical stimuli[22–24], this channel has also been shown to be activated by internal cell-generated forces, which are largely governed by the cytoskeleton[17,47,50]. In addition, changes in the actin cytoskeleton have been observed in response to manipulation of Piezo1, thus suggesting a potential coregulation of the cytoskeleton and ion channel[18,23,50,51]. Given recent reports highlighting the importance of actin in the modulation of macrophage activation[52,53], we next sought to better understand the interplay between actin and Piezo1, and their role in regulating inflammation. We first evaluated changes in actin resulting from Piezo1 modulation by staining with phalloidin, which binds to F-actin. We found that Yoda1-mediated channel activation resulted in significantly enhanced F-actin mean intensity compared to DMSO controls (Fig. 5a). Conversely, *Piezo1ΔLysM* macrophages were observed to have significantly reduced F-actin intensity when compared to *Piezo1fl/+* controls (Fig. 5b). We also evaluated the effect of substrate rigidity and found that cells cultured on stiffer surfaces (20, 40, 280 kPa), which were previously observed to have increased Piezo1 expression, exhibited enhanced F-actin levels—integrated across the cell area, which also changes with stiffness—compared to cells on soft 1 kPa surfaces (Supplementary Fig. 13). These results suggest that Piezo1 activity enhances actin polymerization in macrophages.

We next sought to examine the role of actin in regulating Piezo1-mediated Ca$^{2+}$ influx and macrophage activation. We first exposed BMDMs to latrunculinA (LatA), a potent actin inhibitor, and jasplakinolide (Jasp), an actin stabilizer, and quantified changes in F-actin intensity. As expected, we found that LatA reduced and Jasp enhanced F-actin intensity when compared to DMSO controls (Fig. 5c). We next exposed non-targeting (siControl) and Piezo1 (siPiezo1) siRNA treated Salsa6f+ BMDMs to DMSO, LatA, or Jasp and evaluated for changes in IFNγ/LPS-induced Ca$^{2+}$ activity. Consistent with our previous results, we observed a reduction in Ca$^{2+}$ activity in siPiezo1 when compared to siControl treated cells in the DMSO condition (Fig. 5d–g). We also found that cells exposed to LatA resulted in reduced Ca$^{2+}$ activity in both siControl and siPiezo1 treated BMDMs. In contrast, treatment with Jasp enhanced Ca$^{2+}$ activity in siControl treated cells, which was reduced by siPiezo1 treatment, although the levels were higher than DMSO and LatA treated siPiezo1 cells. These data suggest that inhibition of actin polymerization with LatA reduces IFNγ/LPS induced Ca$^{2+}$ activity, whereas stabilization of actin with Jasp promotes Piezo1 activity and may additionally stimulate other Ca$^{2+}$ channel activity (Fig. 5d–g). We also found that pharmacological inhibition of myosin II using ML7 inhibited Piezo1-mediated Ca$^{2+}$ influx (Supplementary Fig. 14). Together these data suggest a positive feedback regulation, with Piezo1 activity enhancing actin polymerization, which in turn increases Piezo1 activity.

We next examined the role of actin and Piezo1 in modulating inflammatory activation by treating cells with LatA or Jasp and evaluating macrophage response to inflammatory agonists.

*Piezo1ΔLysM* macrophages treated with DMSO generally had reduced IFNγ/LPS mediated inflammatory gene expression when compared to *Piezo1fl/+* DMSO controls, consistent with our earlier results (Fig. 5h–j). We also observed that cells treated with LatA had suppressed inflammatory activation in *Piezo1fl/+* macrophages, which was similar to the behavior of *Piezo1ΔLysM* macrophages treated with DMSO, and further knock out of Piezo1 had no effect (Fig. 5h). In contrast, *Piezo1ΔLysM* and *Piezo1fl/+* BMDMs treated with Jasp had enhanced expression of the inflammatory genes, *Il6* and *Il1b*, when compared to DMSO controls. Expression of *Mcp1* was reduced, similar to what was previously observed with Yoda1 treatment (Fig. 5h–j). Together, these studies confirm the role of actin in regulating macrophage activation, as has previously been observed[52,53], and also suggest a potential for positive feedback between Piezo1 and the actin cytoskeleton which, in turn, enhances macrophage inflammatory activation.

## Discussion

Macrophages encounter environments of varied stiffnesses as they are recruited to tissues throughout the body to facilitate inflammatory and healing responses after injury or infection. In this study, we identify a role for the mechanically activated ion channel Piezo1 in macrophage stiffness sensing, and their responses to inflammatory and wound healing agonists. We find that Piezo1 is highly expressed in macrophages derived from bone marrow, and its activity enhances IFNγ/LPS and suppresses IL4/IL13-induced activation through increased NFκB and decreased STAT6 activation, respectively (Fig. 6). Opposing regulation of inflammatory and wound healing states have also been reported for other transcription factors, KLF4 and KLF6, as well as signaling molecules, AKT1 and AKT2[54–56], which upregulate the expression of inflammatory molecules that simultaneously suppress healing activation, such as TNFα[54,56,57]. While our data suggest that manipulation of Piezo1 itself has no effect on unstimulated macrophages, activation or deletion of this ion channel may indeed regulate molecules that are capable of switching macrophage phenotypes. We also show that Piezo1 regulates the activation of resident peritoneal macrophages. While continuous mechanical stimuli present in vivo are thought to desensitize Piezo1 in resident alveolar macrophages[27], the lung and peritoneum clearly have distinct mechanical environments, which may differentially modulate Piezo1 activity. In addition, culture of cells in vitro may potentially cause loss of mechanical memory in resident macrophage populations and mask the effects of desensitization. Nevertheless, our studies suggest a major role for Piezo1 in modulating the functional responses of broad macrophage populations.

Ca$^{2+}$ plays a pivotal role in regulating essential macrophage functions including inflammatory activation[25,26]. Using Salsa6f-expressing macrophages, we show Ca$^{2+}$ influx in response to the chemical Piezo1 agonist Yoda1, IFNγ/LPS stimulation, and to an extent IL4/IL13 stimulation. While the response of Piezo1 to chemical agonist Yoda1 has been known for some time[44], the relevance of chemical activation of Piezo1 under physiological

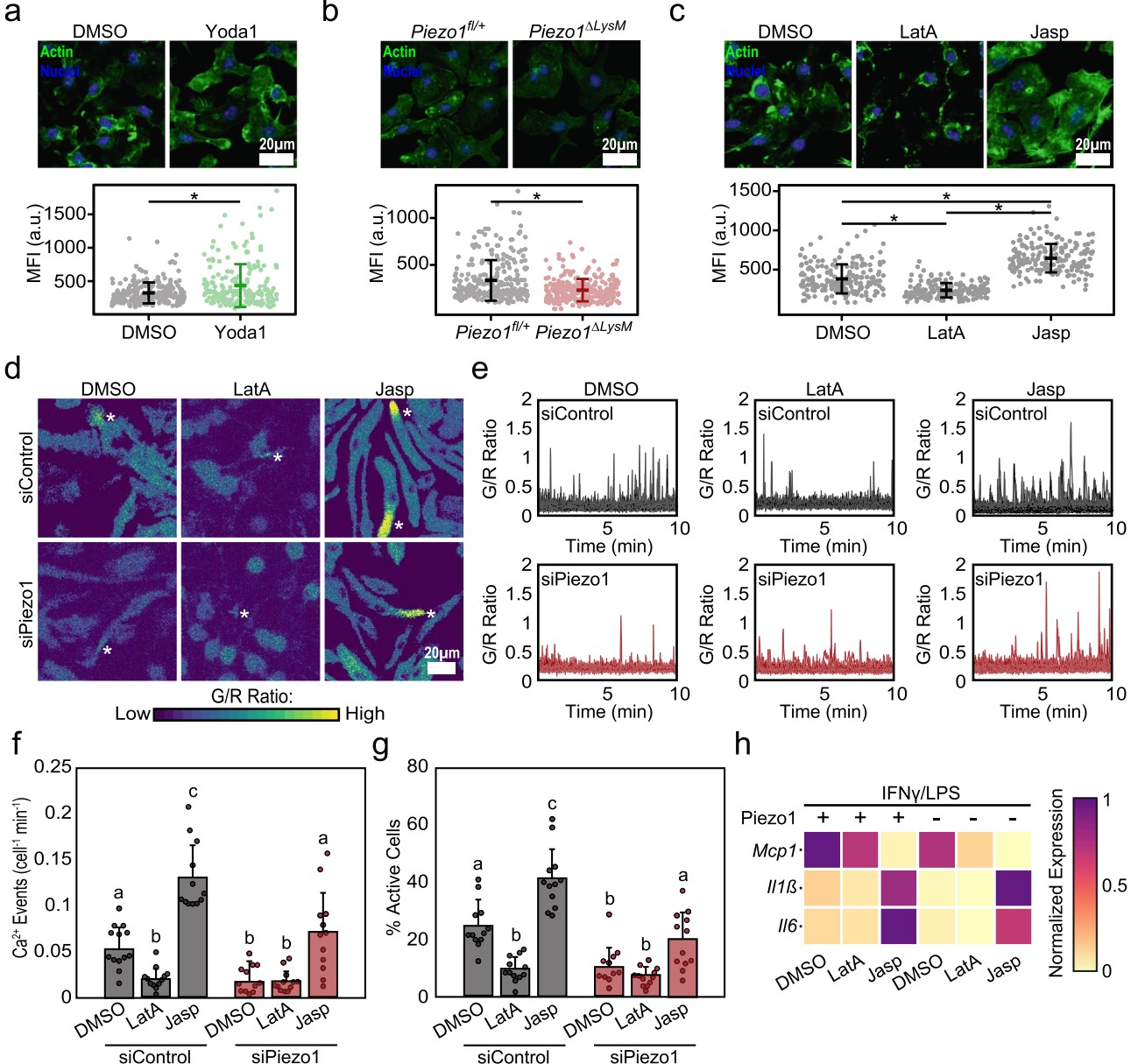

**Fig. 5 Piezo1-mediated regulation of actin influences macrophage inflammatory activation. a** Representative images (top) and quantification (bottom) of actin mean fluorescent intensity (MFI) of wild-type BMDMs following 1 h treatment with DMSO or Yoda1. $N = 229$ and 201 cells examined over three independent experiments for DMSO and Yoda1 treatment. **b** Representative images (top) and quantification (bottom) of actin MFI of *Piezo1*$^{fl/+}$ and *Piezo1*$^{\Delta LysM}$ BMDMs. $N = 277$ and 279 cells examined over three independent experiments for *Piezo1*$^{fl/+}$ and *Piezo1*$^{\Delta LysM}$ BMDMs. **c** Representative images (top) and quantification (bottom) of actin MFI in BMDMs following one-hour treatment with DMSO, 500 nM latrunculinA (LatA), or 500 nM jasplakinolide (Jasp). $N = 195$, 177, and 191 cells examined over three independent experiments for DMSO, LatA, and Jasp treatment. **d–g** Representative G/R ratio images (**d**), traces of individual $Ca^{2+}$ events (**e**), and quantification of number of $Ca^{2+}$ events (normalized for cell number and time) and fraction of cells showing $Ca^{2+}$ elevations, (**f**, **g**) taken from a time-lapse video of siControl and siPiezo1 treated Salsa6f+ BMDMs following addition of DMSO, LatA, or Jasp in Ringer solution containing 100 ng/mL IFNγ/LPS. Asterisks denote the occurrence of a $Ca^{2+}$ event. Each data point in (**f**, **g**) denotes a single video ($N = 12$ videos). **h** Relative *Il6*, *Il1b*, and *Mcp1* gene expression in *Piezo1*$^{fl/+}$ and *Piezo1*$^{\Delta LysM}$ BMDMs exposed to DMSO, LatA, or Jasp and stimulated with IFNγ/LPS for 6 hrs. Gene expression is shown relative to the highest expressing condition. For (**a–c**), error bars denote Mean ± SD, * $p < 0.05$ as determined by two-tailed Mann–Whitney U test. For (**f**, **g**), error bars denote Mean ± SD for $n = 3$, groups not connected by the same letter are statistically different ($p < 0.05$) as determined by two-tailed Student's t test. Source data including exact p-values are provided as a Source Data file.

conditions remains unknown. We propose that soluble cues, such as IFNγ/LPS, may prime the channel's response to mechanical cues, thus regulating the downstream responses to mechanical signals in different contexts. It is possible that $Ca^{2+}$ signals could also arise by store operated $Ca^{2+}$ entry (SOCE) through $Ca^{2+}$ release-activated $Ca^{2+}$ (CRAC) channels, which regulate

signaling pathways essential for T-cell activation[36], though these channels have been shown to have a minimal role in many macrophage functions[37]. Our observations that Piezo1 potentiates NFκB and mitigates STAT6 is consistent with reports that show increased intracellular $Ca^2$ activates NFκB and suppresses STAT6 in immune cells through $Ca^{2+}$-dependent molecules such

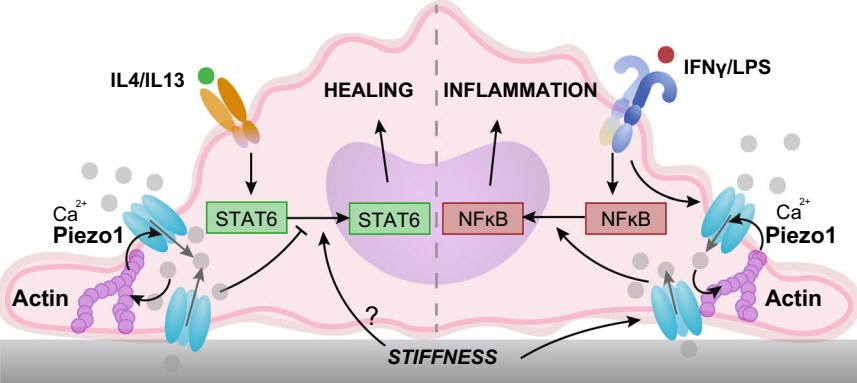

**Fig. 6 Stiffness-dependent modulation of Piezo1 activity modulates macrophage activation.** Activation of Piezo1 by IFNγ/LPS on stiff substrates promotes actin polymerization, which enhances channel mediated $Ca^{2+}$ influx. Positive regulation between actin and Piezo1 enhances inflammation through activation of the transcription factor, NFκB and upregulation of inflammatory markers, including iNOS, TNFα and IL6. $Ca^{2+}$ influx through Piezo1 channels on stiff substrates inhibits the activation of the transcription factor STAT6 and expression of healing markers, such as ARG1. Other stiffness mediated and Piezo1 independent mechanisms likely upregulate STAT6 activation in vitro.

as calpains[25,43]. Furthermore, we show that pharmacological inhibition of calpains reduces inflammatory activation in macrophages, as has previously been shown by others[48,49]. Therefore, Piezo1-mediated $Ca^{2+}$ influx could potentially activate calpains resulting in enhanced inflammatory and suppressed healing activation phenotypes in macrophages.

We also show that Piezo1 plays a role in sensing environmental stiffness in macrophages. Stiff substrates have been shown to promote macrophage inflammatory activation through MyD88-dependent pathways and enhanced NFκB activation[9,58], but the role of stiffness in regulating healing activation has been less clear. We demonstrate stiffness-dependent increases in $Ca^{2+}$ influx in response to the Piezo1 agonist Yoda1 and IFNγ/LPS stimulation. We found that activation of both inflammatory and healing pathways increased as substrate stiffness increased, suggesting that stiffness does not enhance inflammation at the expense of healing, but instead promotes overall macrophage responses to soluble cues. In addition, while inflammation scaled with stiffness, healing responses did not. This could potentially be attributed to the fact that macrophages cultured on 20 and 280 kPa surfaces have enhanced cell spreading and healing activation compared to cells cultured on 1 and 40 kPa surfaces, given that cell shape is known to play an important role in regulating macrophage activation[34,52]. Moreover, stiffness-dependent increases in inflammatory activation required Piezo1, but depletion of Piezo1 reduced inflammation while enhancing healing pathways. Piezo1-mediated $Ca^{2+}$ influx in cells cultured on stiff substrates inhibits IL4/IL13 induced activation of the transcription factor STAT6, but other stiffness-mediated, Piezo1 independent mechanisms could influence STAT6 activation (Fig. 6). Pharmacological inhibition of STAT6 in cells seeded on 1 and 280 kPa substrates revealed suppression of healing activation in cells cultured on both stiffness surfaces, which suggests that STAT6 signaling is prominent even on soft surfaces. Together, our data show that Piezo1-mediated $Ca^{2+}$ signaling promotes inflammation in stiff environments, but the regulation of healing pathways by stiffness and Piezo1 may be more complex.

The activity of mechanically activated ion channels such as Piezo1 has been shown to be dependent on external mechanical cues as well as internal cell-generated forces, and the cytoskeleton is pivotal in transducing such mechanical stimuli as well as generating membrane tension required for channel activity[17,47,59]. We show that positive feedback regulation between Piezo1 and the actin cytoskeleton promotes macrophage inflammatory activation. Specifically, we find that Piezo1

enhances F-actin formation, consistent with what has been observed in lymphatic endothelial cells[51], and that actin polymerization promotes Piezo1-mediated $Ca^{2+}$ activity, suggesting a potential positive feedback regulation between ion channel activity and the cytoskeleton (Fig. 6). Our results also show that inhibition of F-actin with LatA suppresses inflammation, and stabilization of F-actin with Jasp enhances inflammation, consistent with what has been observed by others[52,53]. In addition, Jasp likely enhances other $Ca^{2+}$ sources resulting in increased inflammatory responses in both control and macrophages lacking Piezo1. Actin is known to regulate several TRP channels which could potentially explain this increase in non-Piezo1 mediated $Ca^{2+}$ influx and enhanced inflammation from Jasp treatment[60]. Together, our results suggest that positive regulation between actin and Piezo1 enhances inflammatory activation in macrophages.

Mechanical cues are altered in the context of surgical implants, as well as during the development and progression of many pathological conditions in which macrophages are involved, including various forms of cancer, cardiovascular diseases, and fibrosis[10,13,14]. Our implant studies reveal that Piezo1 is required for macrophage sensing of different stiffness environments in vivo, and that its activity promotes inflammation and suppresses healing function of macrophages. The biomaterials used here allowed comparison of the immune response between soft materials that are similar in stiffness to tissues and the response to considerably stiffer materials that could represent biomedical implants[11]. Heightened stiffness has been associated with increased inflammatory activation in cardiovascular disease and healing activation in cancer and fibrosis[14,61–64], and the more subtle changes in tissue stiffness in such contexts as well as disease models will need to be examined in future studies. Nonetheless, our study identifies a role for the mechanosensitive ion channel Piezo1 in sensing environmental stiffness in myeloid cells and influencing the foreign body response to implanted materials. Further exploring the role of Piezo1 in the context of disease will improve our understanding of the development of various pathological states and the role of macrophages in health and disease.

## Methods
**Animals.** Generation of *Piezo1^{ΔLysM}* mice was accomplished through breeding Piezo1^{flox/flox} (Jackson Laboratories stock no. 029213) and LysM^{Cre/Cre} (Jackson Laboratories stock no. 004781) mice together to generate progeny that were heterozygous for both genes. The generated heterozygotes were then bred with

Piezo1$^{flox/flox}$ mice to generate Piezo1$^{flox/flox}$LysM$^{Cre/+}$ (*Piezo1$^{\Delta LysM}$*) and Piezo1$^{flox/+}$LysM$^{Cre/+}$ (*Piezo1$^{fl/+}$*) mice. A similar breeding scheme was utilized to generate LSL-Salsa6f-Vav1$^{Cre/+}$ mice used for calcium imaging. *Piezo1$^{P1-tdT}$* mice (Jackson Laboratories stock no. 029214) were obtained from the Pathak laboratory. Wild type C57BL/6 J mice (Jackson Laboratories) were also used in experiments that did not require genetic manipulation. Experiments were performed using animals of similar age and littermates were used as controls. Mice were maintained at 12 h light/dark cycles within temperature (70–74 °F) and humidity (30–70%) controlled rooms. All animal experiments were performed in compliance with the University of California, Irvine's Institutional Animal Care and Use Committee under protocol # AUP-20-047.

**Cell isolation and culture**. Bone-marrow-derived macrophages (BMDMs) were harvested from the femurs of 6–12-week-old C57BL/6 J mice. Bone marrow cells were collected by flushing the bone marrow of the femur with DMEM supplemented with 10% heat-inactivated FBS, 2mM L-glutamine, 1% penicillin/streptomyocin (all from Thermo Fisher), and a 10% conditioned media produced from CMG 14–12 cells expressing recombinant mouse macrophage colony stimulating factor (MCSF), which induces differentiation of bone marrow cells to macrophages. To remove red blood cells, the collected bone marrow cells were treated with a red cell lysis buffer, and then centrifuged before being resuspended in the previously mentioned media. After 7 days, the cells were harvested using an enzyme-free dissociation buffer (Fisher Scientific) and seeded onto surfaces that were coated with a 10 μg/mL fibronectin (Corning) solution. BMDMs were seeded at a density of $\sim 3.9 \times 10^4$ cells/cm$^2$ and were incubated overnight prior to stimulation with media (Unstim.), 0.3 ng/mL IFNγ/LPS or 0.1 ng/mL IL4/IL13. Following stimulation, cells were incubated for 1 or 18 h prior to collection. Peritoneal macrophages were harvested following established protocols[65]. Briefly, cold PBS was injected and retrieved from the peritoneal cavity of mice. The resulting cell suspension was centrifuged and resuspended in DMEM supplemented with 10% heat-inactivated FBS, 2mM L-glutamine, 1% penicillin/streptomyocin (all from Thermo Fisher). For studies involving pharmacological manipulation of the cytoskeleton, cells were incubated with either DMSO, 500 nM latrunculinA (LatA), 500 nM jasplakinolide (Jasp), or 25 μM ML7 for an hour prior to stimulation with 0.3 ng/mL IFNγ/LPS for an additional 6 hrs.

**Polyacrylamide gel fabrication**. Polyacrylamide hydrogels of varying stiffness were fabricated using protocols previously described[66]. Briefly, cover slips were cleaned with 70% ethanol and dried prior to 10 min UVO treatment. The cover slips were then treated with bind-silane (solution containing 95 of 95% ethanol, 0.3% 3-(Trimethoxysilyl) propylmethacrylate, and 5 of 10% acetic acid) and were incubated for 5 mins at room temperature. The coverslips were then washed with ethanol prior to incubation at 70 °C for one hour. Meanwhile, glass slides were treated with silanization solution I and incubated in a vacuum desiccator for 5 mins. Glass slides were washed with DI water and blotted dry prior to gel formation. Solutions containing varying ratios of acrylamide:bis-acrylamide were pipetted onto the glass slides and the coverslip was placed onto the solution such that the polyacrylamide gel would be sandwiched between the bind-silane treated coverslip and silanization solution I treated glass slide. Gels were allowed to polymerize for 30 mins and were then removed from the glass slide and placed into culture plates. The resulting hydrogels were conjugated with 20 μg/ml of fibronectin using sulfo-SANPAH (Thermo scientific) overnight at 4 °C.

**Ca$^{2+}$ imaging and analysis**
*Confocal Imaging*. BMDMs from LSL-Salsa6f-Vav1$^{Cre/+}$ mice were seeded on fibronectin coated 35 mm MatTek dishes. Confocal imaging of Ca$^{2+}$ dynamics in Salsa6f macrophages was accomplished using an Olympus Fluoview FV3000RS confocal laser scanning microscope which is equipped with a high-speed resonance scanner and IX3-ZDC2 Z-drift compensator. Cells were maintained at 37 °C using the Tokai Hit incubation stage, excited using sequential line scan at 488 nm and 561 nm and imaged using an Olympus 40x silicone oil objective (NA 1.25). Ratiometric analysis of Ca$^{2+}$ signals was performed through using ImageJ software. Briefly, Green (GCaMP6f) and Red (tdTomato) channel time-lapse images were converted to tiff files and background subtracted. ROIs were drawn around whole cells or individual Ca$^{2+}$ events were outlined, and mean pixel intensities were obtained over time for both the Ca$^{2+}$ independent tdTomato and the green Ca$^{2+}$ sensitive GCaMP6f. Of note, ROIs were drawn around whole cells to quantify responses to Yoda1, which produced global increases in cytosolic Ca$^{2+}$ at the doses used in our study. In contrast, BMDMs exhibited Ca$^{2+}$ events that were often restricted to cell processes or specific regions of the cell at baseline and after the addition of soluble signals, which necessitated drawing of ROIs around individual events for the purpose of quantification. Finally, a ratio of GCaMP6f to tdTomato mean intensities (G/R ratio) was calculated for each ROI and time point to generate Ca$^{2+}$ traces. Peak intensities were obtained through finding the maximum G/R ratio over time across individual cell or ROI. For quantification of responses to soluble signals, individual Ca$^{2+}$ events were outlined as described above. The resulting G/R ratios were then obtained, and the number of events computed through the use of a MATLAB script. Briefly, a polynomial fit was used to compute and subtract baseline values from G/R ratios over time. This was followed by using

a Gaussian filter to smooth the data before using a threshold-based analysis to identify the number of peaks within a signal, which corresponds to the number of Ca$^{2+}$ events. The total number of events was normalized to the number of cells and time of acquisition. The total % active cells was obtained by dividing cells with one or more Ca$^{2+}$ events to the total number of cells present in the field of view.

*TIRF imaging*. BMDMs from LSL-Salsa6f-Vav1$^{Cre/+}$ mice were seeded on fibronectin coated 35 mm MatTek dishes. Imaging of Ca$^{2+}$ dynamics in Salsa6f+ macrophages was accomplished using an Olympus IX83 microscope that was equipped with an automated 4-line cellTIRF illuminator and a PLAPO 60x oil immersion objective (numerical aperture 1.45). Cells were illuminated with 488 and 561 nm lasers and images were acquired with Hamamatsu Flash 4.0 v2+ scientific CMOS cameras at a 100 Hz frame rate. Analysis of the videos was accomplished using Flika, an open-source image analysis package, as previously mentioned[33].

**RNA interference**. For experiments involving the reduction of PIEZO1 expression, unstimulated macrophages were exposed to non-target and PIEZO1 siRNA (both Dharmacon) in a Nucleofector® solution obtained from a primary cell 4D-Nucleofector® kit (Lonza). Following transfection, the cells were supplemented with warm media before being seeded onto experimental substrates. The transfected cells were allowed to adhere for 72 h prior to stimulation for an additional 18 h.

**Western blotting**. BMDMs were rinsed with PBS before being exposed to a lysis buffer, a combination of RIPA lysis buffer and 1% protease inhibitor (both from Fisher Scientific). The substrates were scraped to release the adhered cells and the lysate was collected. The lysate was spun at 16000 g for 15 min and the supernatant was obtained. The proteins were denatured through the use of a Laemmli buffer supplemented with 5% 2-mercaptoethanol at 95 °C for 10 min before each sample was loaded into a well of a 4–15% mini-PROTEAN$^{TM}$ precast gel (all from Biorad). Gel electrophoresis resulted in the separation of proteins before being transferred onto nitrocellulose membranes using the iBlot dry blotting system (Thermo Fisher Scientific). Following electroblotting, the membranes were blocked using 5% nonfat milk in TBST overnight at 4 °C. After 30 min of washing in TBST, the membranes were probed with a primary antibody (see Supplementary table 1 for list of antibodies used) for 1 h at room temperature. An additional 30 min of washing in TBST followed before the membranes were probed with secondary antibodies at room temperature for 1 h. The membrane was then washed in TBST and immersed into a chemiluminescent HRP substrate solution (Thermo Scientific) and imaged using a ChemiDoc XRS System (Biorad). Uncropped blots are provided in the Source Data file.

**ELISA**. Following 18 h of stimulation, the supernatants were collected and analyzed for the presence of TNF-α, IL-6, and MCP-1 using ELISA kits (BioLegend). The assays were conducted following the manufacturer's instructions.

**RNA isolation and qPCR**. Tri-reagent (Sigma) was added to samples to lyse cells and RNA isolation was performed following the manufacturer's instructions. cDNA was made using a cDNA reverse transcription kit (Applied Biosciences) and qPCR was performed using PerfeCTa® SYBR® Green SuperMix (QuantaBio), see Supplementary Table 2 for a list of primers used. All assays were performed in accordance with manufacturer's instructions.

**Immunofluorescence**. Following stimulation, BMDMs were fixed in 4% paraformaldehyde for a period of 10 mins. The fixed cells were washed in PBS prior to permeabilization in 0.1% or 0.3% Triton-X in PBS for staining of PIEZO1$^{P1-tdT}$ or NFκB/STAT6, respectively. Following additional PBS washes the cells were blocked in 2% BSA prior to being incubated with primary antibodies for 1 h at room temperature or overnight at 4 °C (see Supplementary table 1 for list of antibodies used). The cells were then repeatedly washed with 2% BSA and incubated with secondary antibodies in 2% BSA, for 1 h at room temperature. After repeated washing with 2% BSA, the cells were incubated with Alexa Fluor 488 phalloidin (Fisher Scientific), diluted 1:100 in PBS, and Hoechst (Invitrogen), diluted 1:2000 in PBS, for 30 min at room temperature. The cells were thoroughly washed with PBS, before being mounted onto a glass slide and imaged using a Zeiss LSM700 confocal microscope or an Olympus Fluoview FV3000 confocal laser scanning microscope. Approximately 50 cells in each condition were outlined per experiment and the mean intensity or total intensity was computed for each cell using ImageJ.

**Subcutaneous implant animal studies**. Varying stiffness PEGDA gels were implanted into 6-week-old *Piezo1$^{\Delta LysM}$* and control *Piezo1$^{fl/+}$* mice. Stiff (~140 kPa) and Soft (~1 kPa) PEGDA 400 MW (Polysciences Inc.) were reconstituted in PBS at 50% or 10% w/v, respectively, with 0.005% Irgacure 2959 photoinitiator. 1 mm sheets of hydrogels were cast and crosslinked using UV for 5 min and were cut into disks using 5 mm biopsy punches. Prior to implantation, mice were anesthetized using isoflurane and the dorsal skin was shaved and cleansed using 70% ethanol. Subcutaneous pockets were created on either side of a ~5 mm incision

along the dorsal midline, and soft and stiff PEGDA gels were placed inside, one on each side. The incision was closed using staples. Mice were housed individually after wounding and monitored daily for signs of infection/healing. Following 3 days, mice were sacrificed, and the implants were retrieved with surrounding tissue and mounted in OCT for cryosectioning.

**Immunohistochemistry and histology**. Frozen tissue sections were thawed and allowed to equilibrate to room temperature, fixed in 4% paraformaldehyde for 15 min, and then washed 4 times with PBS. Tissues were permeabilized using 0.2% Triton-x-100 in PBS (Sigma) followed by three washes with 0.1% Tween-20 in PBS (Sigma), before blocking in 1% BSA + 0.1% Tween in PBS for 2 h. Sections were incubated in primary antibody overnight at 4 °C (see Supplementary table 1 for list of antibodies used). They were then washed three times and stained with a secondary antibody for 1 h. Following three more washes slides were mounted with Fluoromount G and imaged at 20x using the Olympus FV3000 laser scanning confocal microscope. Images were analyzed using ImageJ software to quantify percent positive staining. Hoechst staining was used to create a nuclear mask and compute the total number of cells in each frame. F4/80 positively stained cells were counted and divided by the total number of cells to get % F4/80+ cells. In addition, to analyze macrophage-specific phenotypes, the number of cells positively stained for inflammatory and/or healing markers were divided by the total number of F4/80+ cells. H&E and Masson's trichrome staining was performed by the University of California, Irvine Research Services Core Facility and were imaged using a Nikon. Images were analyzed using ImageJ software to quantify the average thickness of the collagen capsule from the material/tissue interface to the start of the muscle layer. Images were also analyzed to quantify the number of cells within a given area to assess immune cell infiltration.

**Statistics and reproducibility**. Data are presented as the mean ± standard deviation across at least three independent experiments. Representative images are accompanied with quantification from a minimum of three independent experiments. Comparisons were performed using a two-tailed Student's $t$ test, two-tailed paired $t$ test, or two-tailed Mann–Whitney $U$ test, as indicated in figure legends, and *$p < 0.05$ was considered significant. Exact p-values are provided in the Source Data file.

**Reporting summary**. Further information on research design is available in the Nature Research Reporting Summary linked to this article.

## Data availability
All relevant data supporting the key findings of this study are available within the article and its Supplementary Information files or from the corresponding author upon reasonable request. Source data are provided with this paper. A reporting summary for this article is available as a Supplementary Information file. Source data are provided with this paper.

## Code availability
Flika, an open-source image processing and analysis package, can be obtained from: https://github.com/flika-org/flika.

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

## Acknowledgements

This work was supported by the National Institutes of Health (NIH) National Institute of Allergy and Infectious Disease (NIAID) Grants R21AI128519-01 and R01AI151301, National Institute of Biomedical Imaging and Bioengineering (NIBIB) Grant R21EB027840-01, and National Institute of Arthritis, Musculoskeletal and Skin Diseases (NIAMS) Grant R21AR077288 to W.F.L.; NIAID R01AI121945 and National Institute of Neurological Disorders and Stroke (NINDS) R01NS14609 to M.D.C.; and NIH Director's Fund DP2AT010376 and NINDS R01NS109810 to M.M.P. H.A. was supported by NIH National Institute T32 Training Grant in Cardiovascular Applied Research and Entrepreneurship (5T32 HL116270-3) and American Heart Association Pre-Doctoral Fellowship (20PRE35200220), J.R.H. was supported by HHMI Gilliam Fellowship for Advanced Study awarded to J.R.H. and M.M.P. (GT11549), R.R.N. was supported by the UC Irvine Medical Scientist Training Program NIH training grant (T32 GM008620-18) and NIH NIAID F30 fellowship (AI142988-01A1). This study was made possible, in part, through access to the Optical Biology Core Facility of the Developmental Biology Center, a shared resource supported by the Cancer Center Support Grant (CA-62203) and Center for Complex Biological Systems Support Grant (GM-076516) at the University of California, Irvine. This work was also made possible, in part, through access to a confocal microscope within the Edwards Lifesciences Center for Advanced Cardiovascular Technology supported by (1S10OD025064-01A1) and the microscope imaging core within the Sue and Bill Gross Stem Cell Research Center at the University of California, Irvine. We would like to acknowledge and thank Dr. Adeela Syed and Allia Fawaz for assistance with confocal microscopy as well as Andrew Phan, Hamid Abuwarda, Jessica Chin, Kevin Jiang, and Andrew Flach for their help with experiments.

## Author contributions

H.A., V.S.M., M.D.C., M.M.P., and W.F.L. conceived the project and designed the experiments. H.A., V.S.M., P.K.V., K.T.B., and H.E.L. designed, performed, and analyzed macrophage functional experiments. H.A., A.J., and S.O. designed, performed, and analyzed confocal Ca2+ imaging studies. H.A. and J.R.H. designed, performed, and analyzed TIRF microscopy studies. H.A. and R.R.N. designed, performed, and analyzed implant studies. H.A., A.J., M.D.C., M.M.P., and W.F.L. wrote the manuscript.

## Competing interests

The authors declare no competing interests.
