## [Peer Review File · Nature Communications]

Reviewers' Comments:

Reviewer #1:

Remarks to the Author:

This manuscript received many constructive comments in the first review. As far as I am concerned, all the major issues were reasonably addressed. I was enthusiastic before and remain so.

Reviewer #2:

Remarks to the Author:

The authors have added significant new experiments to support their claims and have made a reasonable effort to rebut those criticisms that they do not address with new data. It can always be argued about what is "outside the scope" but I think their responses are fair.

Especially useful are the studies of actin and myosin inhibition, the additional of studies with a different cell source, and the emphases on what is new here, since several previous studies have already pointed out the mechanosensitivity of macrophages.

One point that was not clear to me is how the total actin was quantified in supple Fig 11. The large difference in total actin (or is it just F-actin?) with stiffness is larger than seen in previous works. What is the actin normalized to?

Reviewer #3:

Remarks to the Author:

In "Mechanically activated ion channel Piezo1 modulates macrophage polarization and stiffness sensing", Atcha et al. have shown how Piezo-1, a crucial mechano-sensitive ion channel, augments the inflammatory response of macrophages while reducing wound healing. In continuation, using multiple approaches, they have also reported how Piezo-1 downregulation decreases pro-inflammatory activation while increasing the pro-healing phenotype in vitro and in vivo. Further, the authors have shown that Piezo-1 activity is important for Ca²⁺ influx, which is necessary for the inflammatory activation of macrophages. Importantly, they have also shown that Piezo-1 expression is modulated by substrate stiffness. Mechanistically, authors have suggested that Piezo-1 and substrate stiffness influence inflammatory activation by modulating levels of NF-κB and phosphorylation of STAT-6.

While some of the observations related to Piezo-1 and the effect of substrate stiffness on macrophage activation are interesting, authors need to provide regulatory molecular mechanisms by which Piezo-1 drives inflammatory activation. This requires the authors to perform RNA-Seq experiments and downstream analysis. Also, some of the results and the supported experiments need further study to concretely confirm the reported observations. I have listed a few in my comments below. I would accept the paper only if the authors perform those experiments and report their findings. Having said that, I appreciate the authors' efforts in driving the field of macrophage mechanobiology.

Comments:

- 1) I am not entirely convinced by the author's claim that Piezo-1 expression reduces wound healing. There is no strong data to support this claim. I suggest the author's tone down this claim substantially.
- 2) As the authors have mentioned that there is a positive feedback between piezo-1 and actin, it would be absolutely crucial to check the nuclear levels and activity of the MRTF-A SRF complex. This is one-way authors could provide regulatory molecular mechanisms by which Piezo-1 drives inflammatory activation. It should also be kept in mind that MRTF-A is downstream of NF-κB and thus might be a better molecular candidate to explore.
- 3) I am not entirely sure of the authors to claim that Piezo-1 has an influence on STAT3

phosphorylation. The western blot is not convincing. Is this the best-represented blot the authors have? 4) Also since authors report that substrate stiffness has an influence on NF-KB, STAT1 and STAT3 is it correct to say that substrate stiffness has a pan influence on all the transcription factors involved in LPS induced pro-inflammatory activation? Are there any LPS induced transcription factors not governed/influenced by substrate stiffness?

5) An absolute requirement for this manuscript is to show that indeed p-STAT6 and these other transcription factors bind more to the promoter of pro-inflammatory genes in macrophages on stiff substrates. Authors should accept the fact that a higher accumulation of transcription factors in the nucleus doesn't necessarily mean higher activity also.

6) Further, I would strongly recommend authors to inhibit p-STAT6 by known drugs in macrophages on the stiff surfaces and compare ARG-1 gene expression in BMDMs on softer substrates.

7) Even though authors have hypothesized that calcium through Piezo1 regulates calpains, which in turn helps to activate NF-KB, there is no experimental evidence provided to support this hypothesis. NF-KB is considered as one of the central LPS induced transcription factors and any suggested mechanism of its regulation needs to be supported by experimental data. I am not expecting any major experiment, but any indication supporting their claim would greatly increase the impact of the story and our understanding of the pro-inflammatory process.

8) Based on previously published work by several groups, Fig. 2a-e seems more like control experiments and doesn't provide much new information except establishing Salsa6f as an alternative to traditional dyes. Similarly, Fig. 3a largely proves previously published data. Thus, I would suggest the authors move them to the supplementary information.

We thank the Editor and Reviewers for their comments. Below, we present each comment, followed by our reply in blue. In response to Reviewer comments, we performed new experiments, as illustrated in new Supplementary Figures (9 and 11) and described in the text. In addition, we present a Reply Figure below. Text edits in the manuscript are indicated by highlighting.

- 1) I am not entirely convinced by the author's claim that Piezo-1 expression reduces wound healing. There is no strong data to support this claim. I suggest the author's tone down this claim substantially.

We have toned down this claim (p. 6).

- 2) As the authors have mentioned that there is a positive feedback between piezo-1 and actin, it would be absolutely crucial to check the nuclear levels and activity of the MRTF-A SRF complex. This is one-way authors could provide regulatory molecular mechanisms by which Piezo-1 drives inflammatory activation. It should also be kept in mind that MRTF-A is downstream of NF-KB and thus might be a better molecular candidate to explore.

Our main objective was to characterize the role of Piezo1 in regulating canonical and well-established activation pathways, such as NFκB and STAT6. While MRTF-A may be involved in the context of the observed actin changes, and a role of MRTF-A in regulating macrophage inflammatory responses was recently reported in a single study (Jain and Vogel. 2018 *Nat. Materials*), our data show that MRTF-A levels decrease dramatically as monocytes are differentiated into macrophages; panel A below shows Western blot of MRTF-A in THP-1 cells differentiated with phorbol myristate acetate (PMA) for the indicated time. Moreover, the changes in response to stiffness appear to be only modest; panel B below shows representative immunofluorescence images of MRTF-A, DAPI, and actin in THP-1 cells cultured on 1 and 280 kPa hydrogels. Therefore, we believe that MRTF-A may not be relevant in the context of our study.

- 3) I am not entirely sure of the authors to claim that Piezo-1 has an influence on STAT3 phosphorylation. The western blot is not convincing. Is this the best-represented blot the authors have?

The quality of the Western blot was influenced by the lower overall expression of this protein requiring higher exposure of the blot. Nonetheless, the experiment was performed multiple times and statistical significance was achieved. We note that STAT3 is often associated with macrophage pro-healing polarization and is activated primarily through IL-10 or IL-6 signaling (Yin et al. 2019 *J BioChem*). We found that IL-10 is not expressed in the stimulation conditions used, and IL6 was expressed only in the IFN γ /LPS stimulation condition (data not shown). Moreover, macrophages deficient in Piezo1 have significantly reduced IL6 secretion and gene expression (**Figure 1b-c**), further supporting our findings that cells lacking Piezo1 have reduced STAT3 activation (**Supplementary Figure 5**). These supplementary data are not the main focus of our work and we believe that further data on STAT3 will not impact the overall conclusions made in our study.

- 4) Also since authors report that substrate stiffness has an influence on NF-KB, STAT1 and STAT3 is it correct to say that substrate stiffness has a pan influence on all the transcription factors involved in LPS induced pro-inflammatory activation? Are there any LPS induced transcription factors not governed/influenced by substrate stiffness?

Our stiffness findings primarily concern NF κ B and STAT6 transcription factors only, and not STAT1 and STAT3 as suggested by the Reviewer. While the question of which transcription factors are regulated by stiffness is interesting, we believe it is out of the scope of this current work. Nonetheless, it has been reported that stiffness does not play a role in regulating MyD88-independent inflammatory pathways (Previtera et al. 2015 PLOS ONE, citation in **line 332**).

- 5) An absolute requirement for this manuscript is to show that indeed p-STAT6 and these other transcription factors bind more to the promoter of pro-inflammatory genes in macrophages on stiff substrates. Authors should accept the fact that a higher accumulation of transcription factors in the nucleus doesn't necessarily mean higher activity also.

We agree with the Reviewer's claim that higher accumulation of transcription factors in the nucleus does not necessarily mean higher activity, and in addition to localization studies we have also shown changes in the phosphorylation states of each transcription factor (**Figure 1d-e**). Given these are canonical pathways, several studies have already reported that enhanced nuclear localization or phosphorylation of NF κ B and STAT6 correspond to enhanced binding to promoters of inflammatory genes (Yu et al. 2019 *Nat. Commun*; Czimmerer et al. 2018 *Immunity*; Hoeksema et al. 2015 *J Immunol*). Several of these references have been added in **lines 80-83**. Moreover, we respectfully point out that several recent studies have relied on changes in phosphorylation states of these transcription factors alone as evidence of enhanced activity (Eyre et al. 2019 *Nat. Commun*; Segatto et al. 2020 *Nat. Commun*).

- 6) Further, I would strongly recommend authors to inhibit p-STAT6 by known drugs in macrophages on the stiff surfaces and compare ARG-1 gene expression in BMDMs on softer substrates.

We thank the reviewer for this suggestion. We have performed additional experiments adding the STAT6 inhibitor, AS1517499, and found that Arg1 expression in response to IL4/IL13 was dramatically inhibited on both stiff and soft surfaces (**new Supplementary Figure 9**). Interestingly, the inhibition of STAT6 reduced Arg1 expression in cells seeded on both soft and stiff surfaces suggests that STAT6 signaling is still present even in soft environments. Our results are consistent with reports in the literature showing STAT6 inhibitors reduce Arg1 in response to IL4 and/or IL13 stimulation in stiff environments (Binnemars-Postma et al. 2018 FASEB J, Cai et al. 2019 JCI Insight). These references have also been provided in **lines 80-83**.

- 7) Even though authors have hypothesized that calcium through Piezo1 regulates calpains, which in turn helps to activate NF-KB, there is no experimental evidence provided to support this hypothesis. NF-KB is considered as one of the central LPS induced transcription factors and any suggested mechanism of its regulation needs to be supported by experimental data. I am not expecting any major experiment, but any indication supporting their claim would greatly increase the impact of the story and our understanding of the pro-inflammatory process.

We thank the Reviewer for this comment. We have now shown that calpain inhibitor I inhibits IFN γ /LPS induced *Nos2* expression as now shown in **new Supplementary Figure 11**, confirming effects of calpain inhibitors that have been reported in several studies (Griscavage et al. 1996 PNAS, Acharya et al. 2019 Front. Immunol). These references are also provided in **lines 276-8**.

- 8) Based on previously published work by several groups, Fig. 2a-e seems more like control experiments and doesn't provide much new information except establishing Salsa6f as an alternative to traditional dyes. Similarly, Fig. 3a largely proves previously published data. Thus, I would suggest the authors move them to the supplementary information.

We believe that the figure panels cited by this Reviewer provide important pretext to the remaining panels associated with each figure. Furthermore, the use of Salsa6f offers unprecedented visualization of calcium signals in macrophages compared to traditional dyes, which become compartmentalized within macrophage vesicles and excluded from the cytoplasm almost immediately after their addition to macrophage cultures. We therefore respectfully disagree with the Reviewer that these data should be moved into the supplement.

REVIEWERS' COMMENTS

Reviewer #2 (Remarks to the Author):

I think the authors have made sufficient improvements to this manuscript, and it will be of interest to a range of readers.

Reviewer #3 (Remarks to the Author):

The authors were only partially responsive to the previous critiques but I am happy to accept the manuscript in the current format.

- 1) I am not entirely convinced by the author's claim that Piezo-1 expression reduces wound healing. There is no strong data to support this claim. I suggest the author's tone down this claim substantially.

We have toned down this claim (p. 6).

- 2) As the authors have mentioned that there is a positive feedback between piezo-1 and actin, it would be absolutely crucial to check the nuclear levels and activity of the MRTF-A SRF complex. This is one-way authors could provide regulatory molecular mechanisms by which Piezo-1 drives inflammatory activation. It should also be kept in mind that MRTF-A is downstream of NF-KB and thus might be a better molecular candidate to explore.

Our main objective was to characterize the role of Piezo1 in regulating canonical and well-established activation pathways, such as NFkB and STAT6. While MRTF-A may be involved in the context of the observed actin changes, and a role of MRTF-A in regulating macrophage inflammatory responses was recently reported in a single study (Jain and Vogel. 2018 *Nat. Materials*), our data show that MRTF-A levels decrease dramatically as monocytes are differentiated into macrophages; panel A below shows Western blot of MRTF-A in THP-1 cells differentiated with phorbol myristate acetate (PMA) for the indicated time. Moreover, the changes in response to stiffness appear to be only modest; panel B below shows representative immunofluorescence images of MRTF-A, DAPI, and actin in THP-1 cells cultured on 1 and 280 kPa hydrogels. Therefore, we believe that MRTF-A may not be relevant in the context of our study.

- 3) I am not entirely sure of the authors to claim that Piezo-1 has an influence on STAT3 phosphorylation. The western blot is not convincing. Is this the best-represented blot the authors have?

The quality of the Western blot was influenced by the lower overall expression of this protein requiring higher exposure of the blot. Nonetheless, the experiment was performed

multiple times and statistical significance was achieved. We note that STAT3 is often associated with macrophage pro-healing polarization and is activated primarily through IL-10 or IL-6 signaling (Yin et al. 2019 *J BioChem*). We found that IL-10 is not expressed in the stimulation conditions used, and IL6 was expressed only in the IFN γ /LPS stimulation condition (data not shown). Moreover, macrophages deficient in Piezo1 have significantly reduced IL6 secretion and gene expression (**Figure 1b-c**), further supporting our findings that cells lacking Piezo1 have reduced STAT3 activation (**Supplementary Figure 5**). These supplementary data are not the main focus of our work and we believe that further data on STAT3 will not impact the overall conclusions made in our study.

- 4) Also since authors report that substrate stiffness has an influence on NF-KB, STAT1 and STAT3 is it correct to say that substrate stiffness has a pan influence on all the transcription factors involved in LPS induced pro-inflammatory activation? Are there any LPS induced transcription factors not governed/influenced by substrate stiffness?

Our stiffness findings primarily concern NF κ B and STAT6 transcription factors only, and not STAT1 and STAT3 as suggested by the Reviewer. While the question of which transcription factors are regulated by stiffness is interesting, we believe it is out of the scope of this current work. Nonetheless, it has been reported that stiffness does not play a role in regulating MyD88-independent inflammatory pathways (Previtera et al. 2015 PLOS ONE, citation in **line 332**).

- 5) An absolute requirement for this manuscript is to show that indeed p-STAT6 and these other transcription factors bind more to the promoter of pro-inflammatory genes in macrophages on stiff substrates. Authors should accept the fact that a higher accumulation of transcription factors in the nucleus doesn't necessarily mean higher activity also.

We agree with the Reviewer's claim that higher accumulation of transcription factors in the nucleus does not necessarily mean higher activity, and in addition to localization studies we have also shown changes in the phosphorylation states of each transcription factor (**Figure 1d-e**). Given these are canonical pathways, several studies have already reported that enhanced nuclear localization or phosphorylation of NF κ B and STAT6 correspond to enhanced binding to promoters of inflammatory genes (Yu et al. 2019 *Nat. Commun*; Czimmerer et al. 2018 *Immunity*; Hoeksema et al. 2015 *J Immunol*). Several of these references have been added in **lines 80-83**. Moreover, we respectfully point out that several recent studies have relied on changes in phosphorylation states of these transcription factors alone as evidence of enhanced activity (Eyre et al. 2019 *Nat. Commun*; Segatto et al. 2020 *Nat. Commun*).

- 6) Further, I would strongly recommend authors to inhibit p-STAT6 by known drugs in macrophages on the stiff surfaces and compare ARG-1 gene expression in BMDMs on softer substrates.

We thank the reviewer for this suggestion. We have performed additional experiments adding the STAT6 inhibitor, AS1517499, and found that Arg1 expression in response to IL4/IL13 was dramatically inhibited on both stiff and soft surfaces (**new Supplementary Figure 9**). Interestingly, the inhibition of STAT6 reduced Arg1 expression in cells seeded on both soft and stiff surfaces suggests that STAT6 signaling is still present even in soft environments. Our results are consistent with reports in the literature showing STAT6 inhibitors reduce Arg1 in response to IL4 and/or IL13 stimulation in stiff environments (Binnemars-Postma et al. 2018 FASEB J, Cai et al. 2019 JCI Insight). These references have also been provided in **lines 80-83**.

- 7) Even though authors have hypothesized that calcium through Piezo1 regulates calpains, which in turn helps to activate NF-KB, there is no experimental evidence provided to support this hypothesis. NF-KB is considered as one of the central LPS induced transcription factors and any suggested mechanism of its regulation needs to be supported by experimental data. I am not expecting any major experiment, but any indication supporting their claim would greatly increase the impact of the story and our understanding of the pro-inflammatory process.

We thank the Reviewer for this comment. We have now shown that calpain inhibitor I inhibits IFN γ /LPS induced *Nos2* expression as now shown in **new Supplementary Figure 11**, confirming effects of calpain inhibitors that have been reported in several studies (Griscavage et al. 1996 PNAS, Acharya et al. 2019 Front. Immunol). These references are also provided in lines **276-8**.

- 8) Based on previously published work by several groups, Fig. 2a-e seems more like control experiments and doesn't provide much new information except establishing Salsa6f as an alternative to traditional dyes. Similarly, Fig. 3a largely proves previously published data. Thus, I would suggest the authors move them to the supplementary information.

We believe that the figure panels cited by this Reviewer provide important pretext to the remaining panels associated with each figure. Furthermore, the use of Salsa6f offers unprecedented visualization of calcium signals in macrophages compared to traditional dyes, which become compartmentalized within macrophage vesicles and excluded from the cytoplasm almost immediately after their addition to macrophage cultures. We therefore respectfully disagree with the Reviewer that these data should be moved into the supplement.